# mRNA Inventory of Extracellular Vesicles from *Ustilago maydis*

**DOI:** 10.3390/jof7070562

**Published:** 2021-07-14

**Authors:** Seomun Kwon, Oliver Rupp, Andreas Brachmann, Christopher Frederik Blum, Anton Kraege, Alexander Goesmann, Michael Feldbrügge

**Affiliations:** 1Institute for Microbiology, Cluster of Excellence on Plant Sciences, Heinrich Heine University Düsseldorf, 40225 Düsseldorf, Germany; Seomun.Kwon@hhu.de (S.K.); Anton.Kraege@hhu.de (A.K.); 2Bioinformatics and Systems Biology, Justus-Liebig-Universität, 35392 Giessen, Germany; oliver.rupp@computational.bio.uni-giessen.de (O.R.); Alexander.Goesmann@computational.bio.uni-giessen.de (A.G.); 3Biocenter of the LMU Munich, Genetics Section, Grosshaderner Str. 2-4, 82152 Planegg-Martinsried, Germany; brachmann@lmu.de; 4Institute for Mathematical Modelling of Biological Systems, Heinrich Heine University Düsseldorf, 40225 Düsseldorf, Germany; Christopher.Blum@hhu.de

**Keywords:** extracellular vesicles (EVs), mRNA, fungal pathogen, plant pathogen, *Ustilago maydis*

## Abstract

Extracellular vesicles (EVs) can transfer diverse RNA cargo for intercellular communication. EV-associated RNAs have been found in diverse fungi and were proposed to be relevant for pathogenesis in animal hosts. In plant-pathogen interactions, small RNAs are exchanged in a cross-kingdom RNAi warfare and EVs were considered to be a delivery mechanism. To extend the search for EV-associated molecules involved in plant-pathogen communication, we have characterised the repertoire of EV-associated mRNAs secreted by the maize smut pathogen, *Ustilago maydis*. For this initial survey, we examined EV-enriched fractions from axenic filamentous cultures that mimic infectious hyphae. EV-associated RNAs were resistant to degradation by RNases and the presence of intact mRNAs was evident. The set of mRNAs enriched inside EVs relative to the fungal cells are functionally distinct from those that are depleted from EVs. mRNAs encoding metabolic enzymes are particularly enriched. Intriguingly, mRNAs of some known effectors and other proteins linked to virulence were also found in EVs. Furthermore, several mRNAs enriched in EVs are also upregulated during infection, suggesting that EV-associated mRNAs may participate in plant-pathogen interactions.

## 1. Introduction

Extracellular vesicles (EVs) are ubiquitously secreted from cells, carrying a diverse array of molecular cargos. The role of EVs in intercellular signalling and communication is particularly interesting, as they can facilitate mass delivery of otherwise intracellular molecules across the extracellular space. EV-associated molecules can induce physiological changes in the recipient cells [1]. In pathogenic microbes, EVs can facilitate both intraspecies coordination of pathogen cells during infection [2], and broader cross-kingdom interaction with host cells [3,4].

Investigations on fungal EVs have identified associated proteins [5], RNAs [6], lipids [7], polysaccharides [8], and metabolites [9]. At the level of individual cells, EVs have been implicated in structural functions such as cell wall remodelling [10] and glucuronoxylomannan capsule formation [8]. At the population level, secretion of EVs in *Candida albicans* is important for biofilm formation and antifungal resistance [11]. Furthermore, EVs of *Cryptococcus gattii* effectuate long-distance coordination of virulence between fungal cells engulfed in different macrophages; EVs from a hypervirulent strain trigger rapid proliferation of less virulent strains in the phagosome [3]. While EVs of some clinically important fungi carry virulence-associated molecules [12] and promote infection [3,13,14], many studies also indicate that fungal EVs stimulate host immune responses to the detriment of the pathogen [15].

The role of EVs in plant-pathogen interaction is not yet well understood, although they have been frequently observed at various plant-fungal interfaces [16,17,18] Biological significance of plant EVs and their cargos have been elucidated in only a few cases. For instance, EVs of the model plant *Arabidopsis thaliana* carry small RNAs (sRNAs) that silence virulence genes in the grey mould fungus *Botrytis cinerea* [19] and the oomycete pathogen *Phytophthora capsici* [20] during infection. *A. thaliana* EVs additionally contain “tiny RNAs” [21] and various defence-related proteins [22]. In another example, sunflower EVs inhibit spore germination and growth of the white mould pathogen, *Sclerotinia sclerotiorum* [23].

EVs of plant pathogenic fungi are only recently being characterised. So far, EV-associated proteomes of the wheat pathogen, *Zymoseptoria tritici* [24], and the cotton pathogen, *Fusarium oxysporum* f. sp. *vasinfectum* (*Fov*; [9]) have been examined. *Fov* secretes EVs with polyketide synthases and a purple pigment. The fractions containing these EVs trigger hypersensitive cell-death in plants, reflecting the necrotrophic lifestyle of this highly prolific mycotoxin producer [9]. While studies on plant pathogen effectors to date have primarily focused on conventionally secreted proteins, such efforts to examine EV cargos could broaden the spectrum of effector candidates, not only to include unconventionally secreted proteins, but also RNAs and metabolites.

Fungal EVs have been found to contain all types of RNA, the majority of the cargo being shorter sRNAs and tRNAs, but also mRNAs and rRNAs [6]. sRNA effectors have been discovered in at least five different filamentous phytopathogens to date [25,26,27,28,29]. These participate in the bidirectional, cross-kingdom RNAi warfare between plants and pathogens [26]. The diversity of RNAs associated with fungal EVs suggest that RNA species other than sRNAs could also be transferred from a pathogenic fungus to function as effectors in host cells. Particularly interesting would be the concept of effector delivery in the form of full-length mRNAs in pathogen EVs. Such mRNAs could theoretically be translated in the recipient host cells to yield multiple proteins and transfer the cost of effector protein production to the host.

*Ustilago maydis* is a biotrophic fungal pathogen of maize [30], which can cause up to 20% yield losses [31]. It is an established model organism for endosome-associated mRNA transport [32] and has secondarily lost the RNAi machinery, so it does not produce canonical sRNAs [33]. This makes it an interesting organism to examine the mRNA cargo of EVs. EV-like structures have long been observed at the interface between *U. maydis* and maize cells during biotrophic infection [16], suggesting their relevance in the interaction. Furthermore, engineered strains are available, where filamentous growth and the concomitant infection program can be induced in axenic culture [34].

In nature, the infectious form of *U. maydis* is the dikaryotic filament, formed by mating of compatible sporidia [30]. Filamentation is brought about by heterodimerisation of complementary bE and bW homeodomain transcription factors from each sporidium, which initiates a transcriptional cascade for infectious development [35,36]. Here we have taken advantage of a laboratory strain, AB33, where complementary bE and bW are both present in the same strain and are inducible by switching the nitrogen source, allowing facile and reproducible filamentation in culture [34]. AB33 induced filaments transcriptionally and developmentally mimic infectious dikaryotic filaments and have been used as a surrogate to study the initial stage of infection. Evidently, many effectors and genes relevant for infection are expressed in AB33 filaments in culture [35,36]. Hence, we have utilised *U. maydis* as an ideal system for an initial survey of EV cargo mRNAs in plant pathogens.

## 2. Materials and Methods

### 2.1. Culture Conditions and EV Isolation

Initial sporidial cultures of strain AB33 from Brachmann et al. [34] were grown to OD_600_ 1.0 ± 0.1 in complete medium [37], supplemented with 1% glucose. The cells were shifted to nitrate minimal medium [37] with 2% glucose (*w*/*v*) to induce filamentation as described previously [34]. Filament cells were pelleted between 15–16 h post induction (hpi) by centrifugation with JA10 rotor (Beckman Coulter, Krefeld, Germany) at 6000 rpm (3951× *g*) for 10 min. Cell pellets were snap-frozen and saved for RNA extraction. The supernatant was passed through 0.45 µm filter (Sarstedt, Nümbrecht, Germany). The filtrate was ultracentrifuged with 45 Ti rotor (Beckman Coulter) at 36,000 rpm (100,000× *g*), 4 °C, for 1 h. Resulting pellet was resuspended in phosphate buffered saline (GIBCO^TM^ PBS; pH 7.2, ThermoFisher, Dreieich, Germany) and treated with PBS (mock), 0.1 µg/µL RNase A (ThermoFisher), or RNase A with 0.1% (*v*/*v*) Triton X-100 (Sigma, Darmstadt, Germany) at 4 °C for 10 min. Protease treatment was not included as incubation at the recommended temperature 37 °C alone compromised sample quality, while protease treatment itself did not produce a qualitative difference. The treated EVs were “washed” by adding PBS and repeating ultracentrifugation. The final pellets were resuspended in PBS and passed through qEVorginal/70 nm size exclusion chromatography columns (IZON, Lyon, France). Fractions enriched in EVs were collected and concentrated with Vivaspin-500 MWCO 1000 kDa concentrator (Sartorius, Göttingen, Germany). EVs were snap-frozen and stored at −80 °C until required.

### 2.2. Microscopy

Grids with EV samples for transmission electron microscopy (TEM) were prepared as previously described with a few modifications [38]. EVs in PBS were placed on 300 sq formvar/carbon grids (Plano, Wetzlar, Germany), fixed with 2% paraformaldehyde (*v*/*v*) in PBS, then 1% glutaraldehyde (*v*/*v*). Samples were contrasted with 4% uranyl acetate (*w*/*v*), 2% methylcellulose (*w*/*v*) and examined with an EM902 transmission electron microscope (Zeiss, Oberkochen, Germany). EVs in PBS were stained with 8 µM FM4-64 (final concentration; ThermoFisher) and examined with a Zeiss Axio Imager M1, equipped with a Spot Pursuit CCD camera (Diagnostic Instruments, Sterling Heights, MI, USA) and Plan Neofluar objective lens (100×, NA 1.3). FM4-64 was excited with an HXP metal halide lamp (LEj) in combination with filter set for mCherry (ET560/40BP, ET585LP, ET630/75BP; Chroma, Bellow Falls, VT, USA). Microscope operation and image processing were conducted with MetaMorph (version 7, Molecular Devices, San Jose, CA, USA). Differential interference contrast images of sporidia and filament cells were obtained with the same instrument with a 63× objective (NA 1.25).

### 2.3. RNA Extraction, Quality Control, and Sequencing

RNA was extracted from EVs and filament samples using standard methods for TRI-reagen LS (Sigma) and TRI-reagent (Sigma), respectively, with a few modifications. Extracted RNA was treated with DNase I (ThermoFisher) as per manufacturer’s instructions and re-extracted with TRI-reagent LS. Coprecipitant GlycoBlue^TM^ (ThermoFisher) was used for EV RNA samples for the first extraction and for all samples in the re-extraction. RNA quality was controlled with Bioanalyzer^TM^ RNA 6000 Nano (Agilent, Santa Clara, CA, USA) assay using the eukaryote setting. Libraries for sequencing were generated with NEBNext^®^ Ultra™ II Directional RNA Library Prep Kit for Illumina together with NEBNext^®^ Poly(A) mRNA Magnetic Isolation Module according to the manufacturer’s instructions (NEB, Frankfurt am Main, Germany). Libraries were quality controlled with High Sensitivity DNA Kit on Bioanalyzer (Agilent) and quantified on Qubit 2.0 Fluorometer (ThermoFisher) with ds HS Assay Kit. Sequencing was performed in the Genomics Service Unit of LMU Biocenter, on Illumina MiSeq with v3 chemistry with 2 × 150 bp paired-end reads (Illumina, San Diego, CA, USA).

### 2.4. Analysis of RNA-seq Data

Raw sequencing reads were quality checked with FastQC (November 2014) [39], adapter sequences and low quality regions (Q20) were trimmed at the end with Trimmomatic (August 2014) [40]. The reads were mapped to the *Ustilago maydis* genome (Umaydis521_2.0, ENSEMBL) [41,42], using HISAT2 (version 2.1.0) [43] with known splice-sites from the ENSEMBL annotation. The library degradation was checked using the geneBodyCoverage.py tool from RSeQC (August 2012) [44] using the BAM file with the mapped reads. Due to short reads, we first merged the paired-end reads using BBMerge (version 2019) [45] and then aligned the merged reads to the reference using HISAT2 [43]. To correct the read counts for potentially degraded transcripts we used the DegNorm tool (version 0.1.4) [46]. The DegNorm-corrected read counts were used for pair-wise differential expression analyses with DESeq2 (version 1.32.0) [47]. Raw reads, DegNorm-corrected counts file, and the DESeq2 results are available at NCBI’s Gene Expression Omnibus [48] (accession number GSE176292).

Principal component analysis on DESeq2 results was visualised with PCAtools (version 2.2.0) [49] and the differentially expressed genes displayed using EnhancedVolcano [50]. Mapped reads were viewed on IGV (version 2.4.10) [51]. A given transcript was considered to have “full CDS coverage” if they meet the following criterion: the entire coding region is covered by at least one read per nucleotide position in at least one out of four biological replicates. GO term and KEGG pathway (version 98.1) [52] overrepresentation analyses were carried out following a published protocol [53], using g:Profiler (version e104_eg51_p15_3922dba) [54] and Cytoscape (version 3.8.2) [55]. On g:Profiler, an ordered gene set analysis was performed, where transcript/gene IDs were sorted with the most enriched in EVs at the top. Default multiple testing correction with g:SCS algorithm was used to test for significance [54]. To test for overrepresentation of KEGG pathways, a custom GMT file created from the KEGG pathway database was used [52]. Transcripts upregulated during infection were defined from a published infection time-course dataset [56] (n = 2316, log2 fold change ≥ 1, padj < 0.01), where a given transcript should be upregulated during at least one infection time-point (0.5–12 days post inoculation; dpi) compared to axenic sporidia, which is the starting inoculum at 0 dpi.

For comparison of 3′ untranslated regions (3′UTRs) of mRNAs enriched in EVs versus those depleted from EVs, UTRs were partially annotated based on the mapped RNA-seq reads. Reads with gaps larger than 10 kb were removed with BBMap (version 38.87 [57]). Reads that extend beyond but still overlap with the exon region of a given gene were selected for UTR annotation with SAMtools (version 1.11) [58]. The UTRs were defined with the following criteria using BEDtools (version 2.29.2) [59]: covered by at least 10 reads per position per sample, in at least three samples. Regions that did not meet these criteria were not annotated. The partially annotated 3′UTRs of transcripts enriched in EVs (n = 655, baseMean > 10, log2 fold change > 1, padj < 0.01) and depleted from EVs (n = 841, baseMean > 10, log2 fold change < −1, padj < 0.01) were compared. Single nucleotide and 4-mer frequencies were calculated for both classes and tested for difference using the normal approximation to the binomial difference.

### 2.5. Validation by RT and RT-qPCR

RNA extracted from RNase-treated EVs and filament cells were cleaned, concentrated with RNA Clean & Concentrator-5 (Zymo Research, Freiburg, Germany). 200 ng of cleaned RNA was used as template for first-strand cDNA synthesis with SuperScript™ IV First-Strand Synthesis System (ThermoFisher), with an inclusive RNase H treatment as according to the manufacturer’s instructions. The first-strand reaction was 8-fold diluted and 1 µL was used as template for PCR with 100 nM primers, following an otherwise standard protocol for Phusion^®^ High-Fidelity DNA Polymerase (NEB) with 35 cycles. Annealing temperature was 60 °C and extension time was 50 s for all reactions. For RT-qPCR, 100 ng of RNA was used as input for first-strand cDNA synthesis. The cDNA was diluted 16-fold and 2 µL was used per reaction in qPCR, following an otherwise standard Luna^®^ Universal qPCR Master Mix (NEB) protocol in Stratagene Mx3000P (Agilent). Relative gene expression analysis was carried out using the 2^^-ddCT^ method [60], with UMAG_02361 as a reference gene between EVs and filament samples (Log2 fold change = −0.09 in RNA-seq; Appendix A). Primers used for full-length RT-PCR and RT-qPCR are shown in Appendix A.

## 3. Results

### 3.1. EVs from Axenic Filamentous Cultures of U. maydis Contain RNA

First, to check whether *U. maydis* hyphae secrete EVs with appreciable RNA cargo, we have developed a robust protocol for EV isolation (or enrichment) from *U. maydis* cultures. EVs were isolated from filaments of strain AB33 [34], induced from yeast-like, budding sporidia (Figure 1a) in axenic culture. Isolated particles were examined by TEM, which confirmed typical cup-shaped form of fixed EVs (Figure 1b). The samples were subjected to staining with the lipophilic dye FM4-64, which further verified the presence of lipid-containing particles (Figure 1c).

In order to determine the presence of extracellular RNA protected within EVs, Bioanalyzer^TM^ profiles of EV-associated RNAs were examined following RNase treatment of EVs prior to RNA extraction. While RNA extracted from EVs treated with RNase alone (Figure 2b) still produced a profile comparable to mock-treated EVs (Figure 2a), RNase treatment in the presence of a detergent (Figure 2c) at a concentration that should disrupt EV membrane integrity [61], led to extensive degradation of EV-associated RNA. This supports that the extracellular RNA isolated is likely to be encased in EVs, protected from the RNase-rich culture environment.

Bioanalyzer^TM^ profiles of EV-associated RNAs showed the presence of distinct 18S and 28S ribosomal RNA (rRNA) peaks and a larger, broader peak of less than 200 nt (Figure 2a,b). 18S and 28S rRNAs occupy a lesser proportion (7.3% ± 1.7; *n* = 4) in EV samples (Figure 2a) compared to total RNA samples of filamentous cells (45.2% ± 3.0; *n* = 4; Figure 2d). Most EV-associated RNA molecules detected were under 200 nt in length. This is in the range of tRNAs and other non-coding RNAs in *U. maydis*, but probably also includes fragmented mRNAs and rRNAs.

Integrity of EV-associated RNAs seems lower, with a mean RNA integrity number (RIN) value of 3.6 ± 1.7, compared to 9.8 ± 0.1 for filament cell RNA (*n* = 4; Figure 2d). This is typical for EV-associated RNAs [62]. The consensus in the EV field is that the RNA integrity number (RIN) provided by the Bioanalyzer^TM^ is not appropriate for RNA from EVs, as shorter RNAs are typically predominant in EVs and relative proportions of different RNA species are likely to be different from total cell RNA [62,63]. The presence of distinct longer rRNA peaks and the absence of notable degradation signals between the major peaks suggest that the higher proportion of shorter RNAs may not simply be attributable to degradation alone, but rather a typical feature of EV-associated RNAs, where shorter transcripts or fragments are more abundant and rRNA is relatively depleted [62,63].

### 3.2. U. maydis EVs Carry a Distinct Pool of mRNAs Compared to Filaments

To create a catalogue of mRNAs in *U. maydis* EVs, sequencing was carried out on poly(A)-enriched libraries of RNA from mock-treated EVs, RNase-treated EVs, and the corresponding hyphal filaments (Figure 2). Reads mapping to rRNA and tRNA regions were also detected, albeit not as abundantly as expected from the Bioanalyser profiles, due to the poly(A)-enrichment method of library preparation. The exact proportions of different RNA species in *U. maydis* EVs remains to be determined.

To assess the variation between all the samples, principal component analysis was carried out following differential expression analysis (Figure 3a). The first principal component (PC1), corresponding to the sample type (EVs vs. filaments), represented 74.7% variance. The EV samples clustered together tightly, regardless of treatment, with their variation no greater than 6.6% (PC2), although the variability was greater among the RNase-treated samples. Mock-treated and RNase-treated EV samples showed a high correlation in read counts (Appendix A). The mean Pearson correlation between the replicates of mock- and RNase-treated EV samples combined is 0.96, while the correlation for replicates from the mock-treated samples alone is 0.97 and RNase-treated samples is 0.96. The mean correlation between EV and filament samples is lower at 0.83.

With the assumption that functionally important mRNA cargo would be selectively loaded and therefore relatively “enriched” inside EVs, differential expression analysis was carried out to identify transcripts differentially associated with EVs compared to filament cells. Transcripts from 1974 out of 6765 protein coding genes were differentially associated with EVs (Figure 3b; Bonferoni-Hochberg adjusted Wald test *p*-value, padj < 0.01, log2 fold change ≥ 1 or ≤−1), of which 758 transcripts were ≥2-fold enriched within EVs and 1189 were depleted from EVs to the same extent (Appendix A). This indicates selective loading instead of random bulk loading of RNA into EVs.

Following the observation that the proportion of shorter RNAs is increased in EVs compared to filaments (Figure 2), we analysed the length distribution of mRNAs in relation to their enrichment within EVs (Figure 3c). This revealed a bias for enrichment of shorter mRNAs; the median for mRNAs relatively enriched in EVs was 1.002 kb, compared to 1.962 kb for depleted transcripts (Wilcoxon rank sum test, W = 775,462, *p*-value = 3.77 × 10^−108^), and 1.523 kb for those in neither category (Wilcoxon rank sum test, W = 1,109,078, *p*-value = 2.26 × 10^−55^; Figure 3c). This is in agreement with the notion that larger size can hinder RNA loading into EVs [64]. In essence, RNA-seq of *U. maydis* EV samples has revealed the presence of thousands of mRNAs associated with EVs and relative enrichment of certain population of mRNAs in EVs compared to filament cells is the first indication that there might be specificity in loading of mRNAs into EVs.

### 3.3. Confirmation of Enriched mRNAs with Full-Length CDS in EVs

If mRNAs in EVs are transferred to recipient cells for a specific biological purpose, they could either be translated into functional proteins and/or be fed into the RNAi machinery to silence gene expression. Full-length mRNAs are prerequisite for the first scenario, while fragments should suffice for the latter. Hence, we have checked for the coverage of coding sequences (CDS) in our RNA-seq experiment. Annotations of untranslated regions (UTRs) are not available for *U. maydis*, but the read coverages continuously extending beyond the CDS indicates that the UTRs may be intact for several transcripts. Over half of all transcripts detected in the RNA-seq experiment had full CDS coverage in EVs (Figure 3d). Furthermore, 92.9% of transcripts significantly enriched in EVs (n = 758, log2 fold change ≥ 1, padj < 0.01), had full CDS coverage, suggesting the presence of full-length mRNAs in EVs (Figure 3d).

For verification, four enriched mRNA candidates, that have previously been shown to be upregulated during infection [56], were chosen to secondarily confirm the presence of full-length CDS and enrichment in EVs. Two candidates that encode putative oxidoreductases, UMAG_02984 and UMAG_04370, were chosen as they were among the most highly enriched mRNAs in EVs (Appendix A). The other two candidates, UMAG_11400 and UMAG_01171, encode metabolic enzymes and were chosen among the less enriched mRNAs in EVs (log2 fold change ~1), in order to test a range of enrichment levels. Presence of full-length transcripts in EVs was checked first by RNA-seq read coverage (Figure 4a) and then by reverse-transcription with oligo-d(T) primers, followed by PCR with primers covering at least 90% of the coding region (Figure 4b). Hence, the intactness of the poly(A) tail and the exon region could be inferred. Relative enrichment of the candidate mRNAs in EVs versus filament cells was checked by RT-qPCR, which was in agreement with the RNA-seq results (Figure 4c). Thus, we have demonstrated that the presence of full-length mRNAs, enriched inside EVs is highly probable, opening up the possibility that fungal mRNAs might be translated in the host.

### 3.4. Functional Enrichment of mRNAs in EVs

With the notion that mRNAs enriched inside EVs are more likely to be functionally important, we carried out GO term overrepresentation analysis on transcripts differentially associated with EVs. Indeed, mRNAs relatively enriched in EVs showed overrepresentation of different functional GO terms from those that are depleted (Figure 5; Appendix A). Transcripts enriched within EVs (log2 fold change ≥ 1, padj < 0.01, baseMean ≥ 10) showed significant overrepresentation of biological process GO terms for various metabolic processes, proteosomal protein degradation, vesicle-mediated transport, organisation of actin filaments, cytokinesis, and pathogenesis (Figure 5a; g:SCS padj < 0.05). Accordingly, overrepresented molecular function GO terms were mostly enzymatic activities or proteasome-related (Figure 5b). Although the GO term for “pathogenesis” was overrepresented (Figure 5a), the enriched mRNAs are mostly involved in iron uptake, and those that have been examined in *U. maydis* seem to play a role in nutrient acquisition rather than having a direct virulence function [65]. Overrepresented cellular compartment GO terms indicated that the protein products of mRNAs enriched in EVs localise to the cytosol, membranes of vacuoles and vesicles, the proteasome, and the septin complex (Figure 5c). Overrepresentation of GO terms related to intracellular vesicle transport and septins might reflect the link between endosomes and EVs [66], and septin mRNAs are confirmed cargos of endosome-associated mRNA transport in *U. maydis* [67,68].

Transcripts that are relatively depleted from EVs (log2 fold change ≤ −1, padj < 0.01) were involved in transmembrane transport, cell wall processes, signal transduction, and several ER-related process such as protein glycosylation, glycolipid metabolism, ER organisation, and the ER-associated protein degradation (ERAD) pathway (Figure 5a). Accordingly, these were predicted to function predominantly at the ER and the plasma membrane (Figure 5c). In agreement with the depletion of ER-targeted mRNAs, transcripts of conventionally secreted proteins were also generally depleted from EVs (n = 1113, log2 fold change ≤ −1, padj < 0.01, baseMean ≥ 10, g:SCS padj = 2.08 × 10^−13^). These results suggest that subcellular localisation of mRNAs may affect their loading into EVs: mRNAs associated with intracellular vesicles are more likely to be loaded into EVs, while those that require translation at the rough ER are relatively depleted from EVs.

Since the 3′UTR region is particularly important for subcellular localisation of mRNAs [69], we have partially annotated the UTRs based on the mapped sequencing reads and carried out a preliminary analysis to test if there is a difference between the 3′UTRs of mRNAs enriched in EVs and those depleted from EVs. 3′UTRs of enriched mRNAs showed a higher frequency of adenine nucleotides (*p* = 1.3 × 10^−19^) and less cytosine (*p* = 1.1 × 10^−7^) and uridine (*p* = 9.6 × 10^−3^) compared to the depleted sequences. Accordingly, A-rich 4-mers were significantly increased in frequency among the enriched 3′UTR sequences compared to the depleted (*p* < 0.05; Appendix A). This prompts deeper investigation into EV-targeting motifs in the future.

### 3.5. mRNAs Upregulated during Infection Are Present in EVs from Axenic Filaments

Since *U. maydis* is a plant pathogen, we asked whether a portion of EV-associated mRNAs are relevant for infection. Many genes pertinent for infection are expressed in AB33 filaments in culture due to the transcriptional cascade instigated by bE/bW [35]. Indeed, mRNAs of at least nine previously characterised *bona fide* effectors and secreted proteins linked to virulence were reliably detected in EVs: Stp2 [70], ApB73 [71], Scp2 [72], UMAG_01690 [73], Sta1 [74], Stp1 [75], Nuc1 [76], Cmu1 [77], and UmFly1 [78] (in the order of enrichment in EVs; Appendix A).

Next, we searched for mRNAs enriched in EVs, that are also upregulated during plant infection. For this, we referred to the published time-course transcriptomic analysis of *U. maydis* infection [56]. 161 mRNAs were found to be both enriched in EVs and upregulated during infection compared to the axenic sporidia at 0 dpi (Figure 6a & Appendix A). Over three-quarters of these were induced early on, during the first four days of infection (Figure 6b). GO term analysis of these 161 mRNAs found an overrepresentation of oxidoreductase and other catalytic enzyme activities, as well as functions linked to sulphur compound catabolism and homocysteine metabolism (g:SCS padj < 0.05; Figure 6c). We further examined KEGG pathways [52] and found significant overrepresentation of functions in metabolic pathways, including beta-alanine metabolism, aromatic amino acid biosynthesis, nitrogen metabolism, and glycerolipid metabolism (g:SCS padj < 0.05; Figure 6d). If pathogen EV-associated mRNAs can act as effectors, such metabolic enzymes may be relevant, as *U. maydis* is known to reprogram plant host metabolism [79].

We have examined the most highly enriched mRNAs (*n* = 17, Log2 fold change ≥ 3, padj < 0.01, baseMean ≥ 10), with the assumption that these are more likely to have been loaded in EVs to serve a biological function (Table 1). Many of the most enriched mRNAs encode oxidoreductases with similar annotations, suggesting related activities. This reflects the general overrepresentation of GO terms for oxidoreductases and metabolic enzymes (Figure 5 and Figure 6). Secondly, 10 out of the 17 most enriched mRNAs are induced concomitant with filamentous growth (Log2 fold change ≥ 1, padj < 0.01; [80]), and are up-regulated during infection (Log2 fold change ≥ 1, padj < 0.01; [56]). Furthermore, with reference to the previously defined co-expression modules from an extensive infectious time-course study [56], we found an overrepresentation of the “magenta” infection-related expression module representative of biotrophic proliferation *in planta* (g:SCS padj = 1.43 × 10^−4^). In essence, we observe an enrichment of mRNAs in EVs that can be linked to filament induction and infection.

## 4. Discussion

EVs are emerging as mediators of plant-pathogen communication, particularly as vehicles for transfer of RNA (reviewed in [81]). On the plant side, studies to date have mostly focused on the role of sRNAs [19,20] and proteins [22,23,82] in EVs, while only protein cargos have been examined in EVs of phytopathogenic fungi [9,24]. To extend the search for EV cargo molecules in plant-pathogen communication, we have characterised the repertoire of mRNAs associated with EVs of the fungus *U. maydis*.

For this purpose, we developed a robust EV isolation (or enrichment) protocol and examined EVs produced in axenic cultures of *U. maydis* filaments, used as a surrogate for infectious hyphae *in planta* (Figure 1). Omics studies on EVs of phytopathogenic fungi have so far examined EVs from axenic cultures [9,24]. While there are limitations to using axenic cultures of pathogenic fungi to identify EV cargos linked to virulence, isolation of fungal EVs from apoplastic washing fluid of maize plants is inherently destructive [83,84] and would first require development of markers for *U. maydis* EVs. Induced filaments of the lab strain AB33 in axenic culture mimic the morphology and, partially, the gene expression of infectious filaments [34,35,36] Therefore, we have used these cultures for an initial survey of EV-associated mRNAs in *U. maydis*.

We have reliably detected transcripts of *bona fide* effectors and several secreted proteins linked to virulence in AB33 filaments and their EVs (Appendix A), which supports that our system has the potential to lead to discovery of novel EV-associated effectors. Thousands of mRNAs were detected in association with *U. maydis* EVs, the majority of which have full-length coverage (Figure 3d and Figure 4) and are protected from external RNases (Figure 2). Protease activity in *U. maydis* cultures is high, requiring deletion of multiple proteases to obtain intact secreted proteins from *U. maydis* cultures [85]. Likewise, *U. maydis* secretes RNases in culture [76]. Therefore, it is unlikely that so many mRNAs can preserve integrity in the culture medium, unless they are protected inside EVs. Over 90% of the mRNAs enriched inside EVs are likely to be full-length (Figure 3d), suggesting that there is a biological reason for loading these mRNAs into EVs.

mRNA loading into EVs may be determined by the intracellular location of the mRNAs inside the fungal filament. The two most relevant EV subtypes are exosomes and microvesicles. Exosomes are originally intraluminal vesicles (ILVs) in multivesicular endosomes (MVEs) that are released upon fusion with the plasma membrane, while microvesicles are formed by direct budding from the plasma membrane [66]. Hence, localisation on the surface of maturing endosomes or at the cell periphery would increase the likelihood of being loaded into exosomes and microvesicles, respectively. This might explain why mRNAs encoding proteins linked to intracellular vesicles and vacuoles are enriched in EVs of *U. maydis* (Figure 5). Discovery of EV-associated RNA-binding proteins should help elucidate the mechanism of mRNA loading.

Since mRNAs enriched within EVs encode proteins with functions distinct from those that are depleted, we suspect a biological reason for preferentially exporting these mRNAs. Since several transcripts upregulated during infection are enriched in *U. maydis* EVs (Table 1; Figure 6a), such mRNAs could be studied further as effector candidates. There are two non-mutually exclusive hypotheses for the role of EV cargo mRNAs in *U. maydis*-maize interaction: (1) fungal mRNA fragments lead to silencing of maize genes or (2) full-length fungal mRNAs are translated into multiple effector proteins in maize cells.

Bidirectional, cross-kingdom RNA interference (RNAi) is a widespread mechanism of plant-pathogen interaction [86]. Diverse fungal and oomycete pathogens send sRNA effectors that hijack the plant RNAi machinery to silence host defence genes [20,25,26,27,28,29] As is the case for plant sRNAs that target pathogen genes [19], EVs are thought to be the vehicles of pathogen sRNA effector delivery to host plant cells. *U. maydis* has lost the conventional RNAi machinery [33]. Therefore, it might employ other RNA species, such as mRNA or tRNA fragments for the same purpose. For example, tRNA-fragments of the bacterial symbiont *Bradyrhizobium japonicum* participate in silencing of plant genes involved in root hair development to promote nodulation [87].

Effector delivery in the form of mRNA could be highly cost-effective for the pathogen, if they can be translated in the correct location at required amplitude in the host cell. In support of this hypothesis, proof of principle studies using elegant reporter systems have demonstrated that EV-associated mRNAs are transferred and translated de novo in the recipient cells [88,89]. A recent in vivo study has shown that mRNAs in glioblastoma EVs are most likely translated in recipient astrocytes and lead to metabolic reprogramming [90]. Also, in the medically important *Paracoccidioides* spp., mRNA cargos of EVs were found to be translation-competent in a heterologous, in vitro system [91]. Given these examples and the evidence for full-length EV cargo mRNAs from this study (Figure 3d and Figure 4b), translation of EV-associated fungal mRNAs into functional proteins in recipient cells seems possible.

It is interesting that the set of mRNAs enriched in EVs and upregulated during infection are overrepresented in metabolic enzymes and oxidoreductases (Figure 6). Biotrophic colonisation by *U. maydis* is accompanied by extensive reprogramming of metabolism, redox status, and hormone signalling in the infected plant tissues [79]. The fungus deploys effectors to divert metabolites away from biosynthesis of lignin [92] and salicylic acid (SA; [77]), and induces the jasmonate/ethylene signalling pathway to counter SA-mediated defence [93]. *U. maydis* also harbours metabolic enzymes lacking signal peptides that can synthesise [94,95], degrade [96], or potentially alter metabolic flux into biosynthesis of plant hormones [41]. Intriguingly, isochorismatases, which divert isochorismate away from SA biosynthesis in the host cell, are unconventionally secreted effectors of the filamentous phytopathogens, *Verticillium dahliae* and *Phytophthora sojae* [97]. Similarly, fungal metabolic and redox enzymes, that are upregulated during infection and loaded into EVs in the form of mRNA or protein, have the potential to “moonlight” as effectors if delivered to the host cell. Thus, the presence of intact, enriched mRNAs in EVs of *U. maydis* present an opportunity to discover novel RNA effectors in plant pathogenic fungi.

## 5. Conclusions

We have isolated EVs from the phytopathogenic fungus *U. maydis* and identified mRNAs that are enriched within EVs compared to the cells. Many of the highly enriched mRNAs are also upregulated during infection and are likely to be full-length. The inventory of these mRNAs now forms the foundation for future research addressing the mechanism of mRNA loading into EVs and their function in a recipient cell.

## Figures and Tables

**Figure 1 jof-07-00562-f001:**
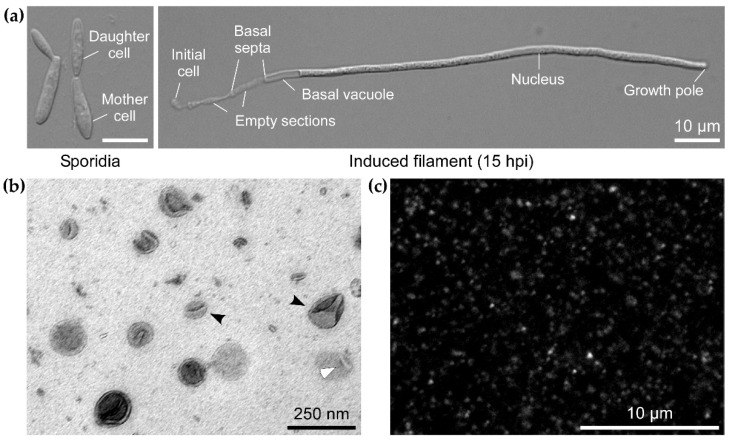
Extracellular vesicles (EVs) from axenic filamentous culture of *Ustilago maydis*. (**a**) Infectious filamentous development was induced in axenic culture using the laboratory strain AB33 [34]. In this strain, the transcriptional cascade of genes necessary for infection and dimorphic switch from yeast-like budding sporidia (left) to hyphal filament (right), can be induced by switching the nitrogen source (both scale bars = 10 µm). EVs were prepared from cultures of filaments between 15–16 h post induction as the one shown on the right. (**b**) Transmission electron micrograph of EVs from filamentous culture of AB33 (scale bar = 250 nm). Typical cup-shaped morphology of EVs is due to fixation. Examples of a smaller and a larger EV are indicated with black arrowheads and an EV lysed during sample preparation is marked with an empty arrowhead. (**c**) Staining of AB33 filament EVs with the lipophilic dye FM4-64 (scale bar = 10 µm). Larger brighter spots are most likely aggregates of EVs formed due to ultracentrifugation.

**Figure 2 jof-07-00562-f002:**
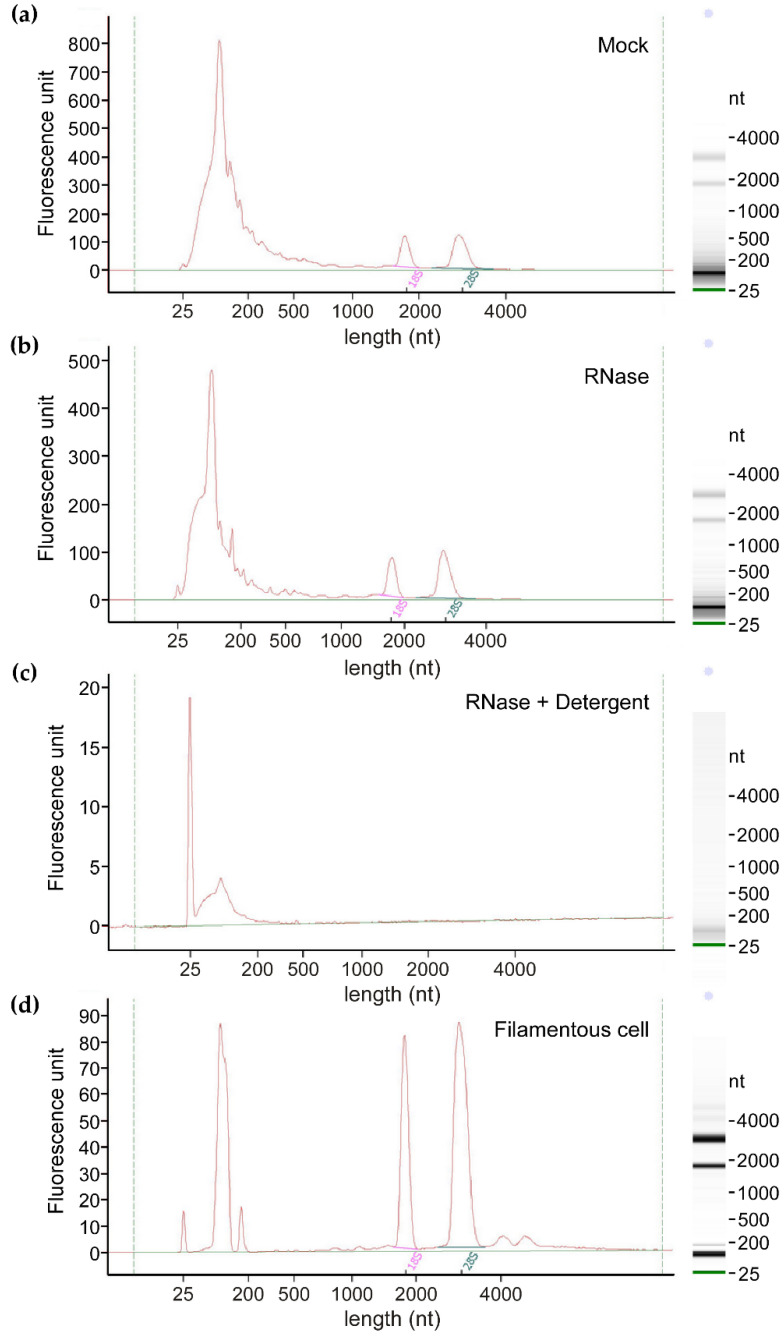
Extracellular RNA associated with *U. maydis* EVs. (**a**) Bioanalyzer profile of RNA extracted from EVs suspended in PBS, incubated with additional PBS as a mock treatment. (**b**) RNA from EVs treated with 0.1 µg/µL RNase A in PBS. (**c**) RNA from EVs treated with 0.1 µg/µL RNase A and 0.1% (*v*/*v*) triton X-100 in PBS. (**d**) Total RNA of induced hyphal filament cells, from which above EV samples were derived.

**Figure 3 jof-07-00562-f003:**
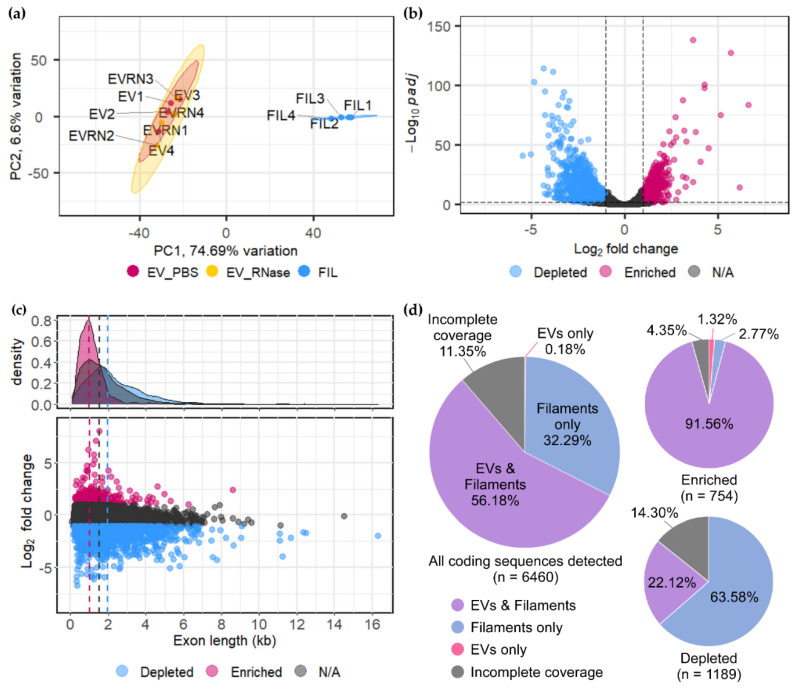
mRNA content of EVs is distinct from that of the hyphal filaments from which they originate. (**a**) Principal component analysis representing “differential expression” or differential presence of mRNAs in four corresponding sets of mock (EV_PBS; red) and RNase-treated (EV_RNase; yellow) EV samples and hyphal filament samples (FIL; blue). (**b**) Volcano plot of transcripts relatively enriched within EVs (red; n = 758, log2 fold change ≥ 1, padj < 0.01) and depleted from EVs (blue; n = 1189, log2 fold change ≤ −1, padj < 0.01) compared to hyphal filaments. (**c**) Effect of transcript length on mRNA enrichment in EVs (log2 fold change). The median length of enriched transcripts is 1.002 kb (red dotted line), is shorter compared to 2.082 kb for depleted transcripts (blue dotted line; Wilcoxon rank sum test, W = 775,462, *p*-value = 3.77 × 10^−108^) and 1.523 kb for those neither enriched nor depleted (grey dotted line; log2 fold change > −1 and <1, Wilcoxon rank sum test, W = 1,109,078, *p*-value = 2.26 × 10^−55^). (**d**) Percentage of transcripts with full read coverage of the coding sequences (CDS); the entire coding region should be covered by at least one read per nucleotide position in at least one out of four biological replicates. Pie charts are shown for all 6460 coding transcripts detected in EVs and for those relatively enriched in EVs and depleted from EVs.

**Figure 4 jof-07-00562-f004:**
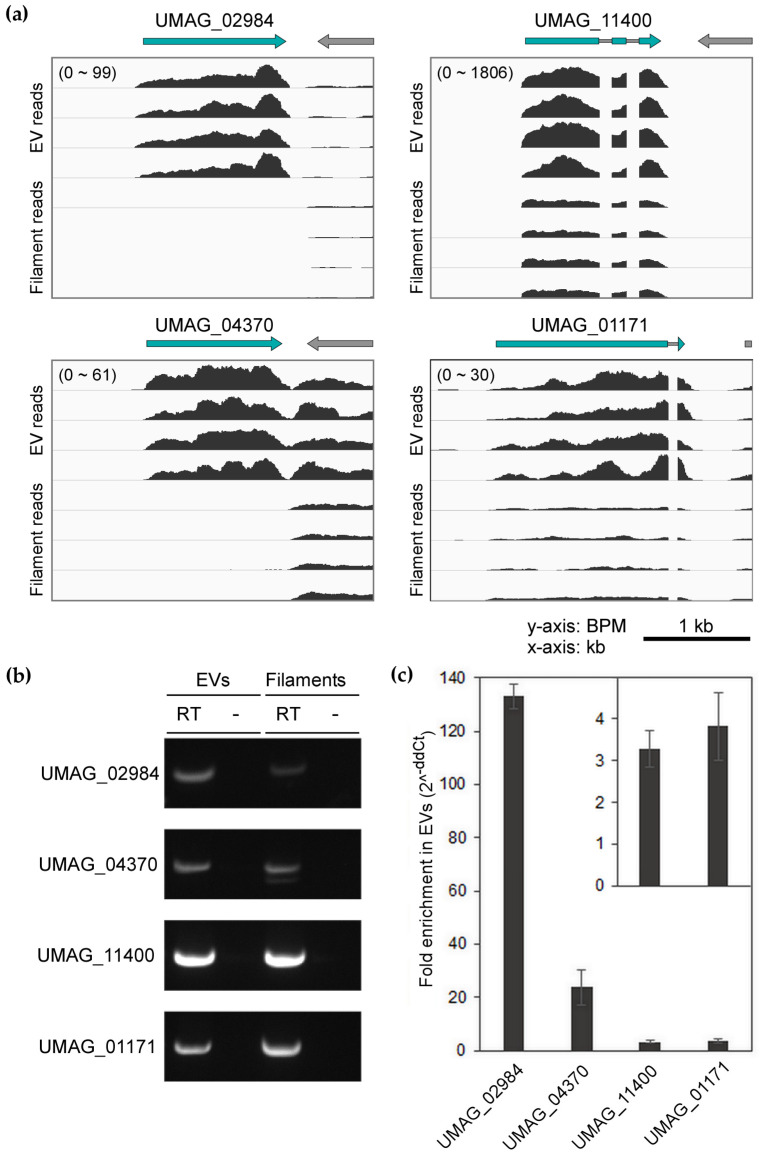
Presence of full-length mRNAs enriched within EVs. (**a**) RNA-seq read coverage of selected infection-relevant, EV-associated mRNA candidates in four biological replicates each of EV and filament samples. *Y*-axis shows normalised coverage in bases per million (bpm) and the range is indicated in brackets. *X*-axis is length in kb (scale bar = 1 kb). (**b**) Confirmation of full-length mRNA candidates by RT-PCR. Primers to yield amplicons covering ≥ 90% of transcript coding region length were used. RT indicates that the reverse-transcribed first-strand cDNA was used as a template for PCR and “-“ sign indicates a -RT negative control. (**c**) Confirmation of relative transcript enrichment in EVs compared to filaments by RT-qPCR. Fold relative enrichment within EVs calculated as 2^-^ddCt^. Inset shows fold enrichment of UMAG_11400 and UMAG_01171 with an adjusted *y*-axis.

**Figure 5 jof-07-00562-f005:**
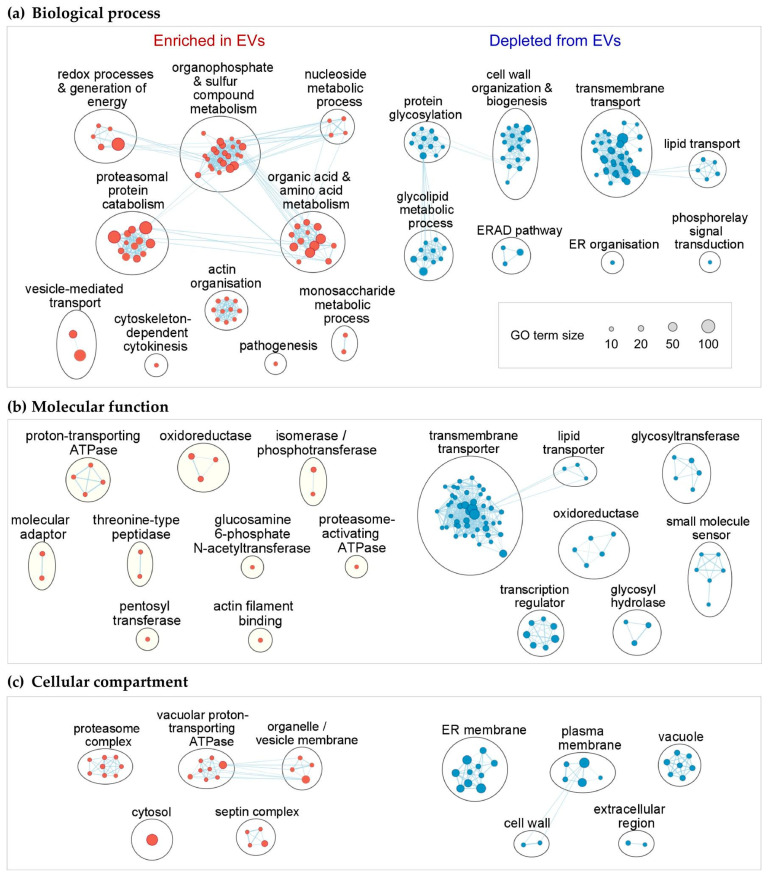
Gene ontology (GO) term analysis of mRNAs differentially loaded into EVs. Biological process (**a**), molecular function (**b**), cellular compartment (**c**). GO terms significantly overrepresented (g:SCS padj < 0.05) in sets of transcripts enriched (red clusters; n = 748, baseMean ≥ 10, log2 fold change ≥ 1, padj < 0.01) and depleted from EVs (blue clusters; n = 1113, baseMean ≥ 10, log2 fold change ≤ −1, padj < 0.01).

**Figure 6 jof-07-00562-f006:**
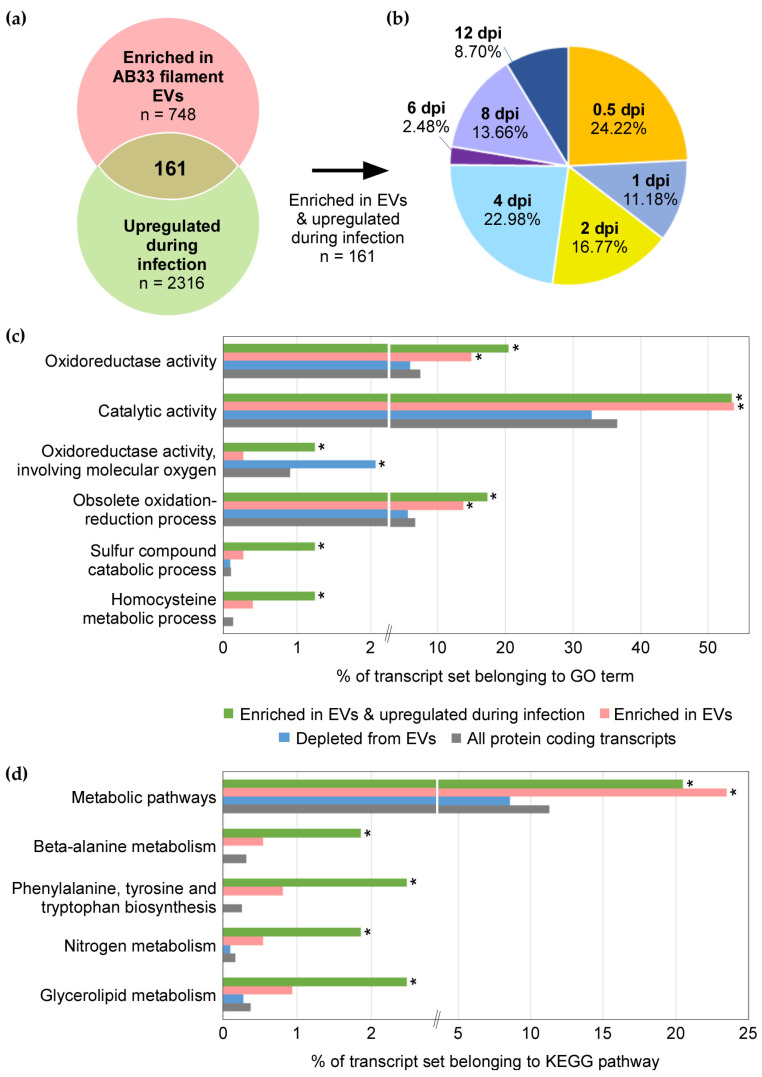
mRNAs enriched within EVs and upregulated during infection. (**a**) Overlap between transcripts enriched within EVs of induced filaments and those upregulated during plant infection. Pink circle represents mRNAs enriched in EVs relative to induced filaments are from this study (n = 748, Log2 fold change ≥ 1, padj < 0.01, baseMean ≥ 10). Green circle represents are mRNAs upregulated in infectious hyphae at 0.5–12 days post inoculation compared to axenic sporidia at 0 dpi (n = 2316, Log2 fold change ≥ 1, padj < 0.01; original data from Lanver et al. [56]). (**b**) Pie chart showing peak expression time-points of 161 mRNAs both upregulated during infection and enriched in EVs. (**c**) GO terms and (**d**) KEGG pathways overrepresented in sets of transcripts enriched in EVs and upregulated in plants (green; n = 161), all enriched in EVs (pink; n = 748, baseMean ≥ 10, Log2 fold change ≥ 1, padj < 0.01), all depleted from EVs (blue; n = 1113, baseMean ≥ 10, Log2 fold change ≤ −1, padj < 0.01), and all protein coding transcripts known in *U. maydis* (grey; n = 6765). Asterisk indicates significant overrepresentation compared to all protein coding transcripts (g:SCS padj < 0.05).

**Table 1 jof-07-00562-t001:** mRNAs most highly enriched in EVs of induced filaments (Log2 fold change ≥ 3, padj < 0.01, baseMean ≥ 10). The 5th column from the left contains values obtained by analysing the raw data from Olgeiser et al. [80]. The 6th and 7th columns contain values from the supplementary dataset published by Lanver et al. [56].

GeneID	Uniprot Annotation	TPM in EVs	Enrichment in EVs vs. Filaments (Log2FC)	Induction in Filaments vs. Sporidia (Log2FC) [80]	Induction during Infection 0.5–12 dpi vs. 0 dpi Sporidia (Largest Log2FC) [56]	Infection Time Course Co-Expression Module [56]
UMAG_02215	flavin-binding monoxygenase	63	8.01	3.31	10.53 (2 dpi)	Magenta (biotrophy)
UMAG_02984	acyl-CoA dehydrogenase	335	7.08	5.70	12.46 (4 dpi)	Magenta (biotrophy)
UMAG_03995	TauD family 2-oxoglutarate- dependent taurine dioxygenase	575	5.78	3.44	8.11 (4 dpi)	Magenta (biotrophy)
UMAG_04370	TauD family 2-oxoglutarate-dependent taurine dioxygenase	256	5.25	3.55	11.21 (2 dpi)	Magenta (biotrophy)
UMAG_06042	2-oxoglutarate/Fe(II)-dependent dioxygenase	185	4.87	4.03	7.71 (4 dpi)	Magenta (biotrophy)
UMAG_00145	serine/threonine protein kinase	827	4.28	0.06	0.54 (12 dpi)	Cyan (tumour)
UMAG_01433	enoyl-CoA isomerase/hydratase fer4 in siderophore ferrichrome A biosynthesis	267	4.26	−0.09	−4.21 (8 dpi)	Burlywood
UMAG_02006	secreted peptidase	498	4.24	4.61	8.29 (1 dpi)	Red (Plant surface)
UMAG_11874	uncharacterised protein	57	4.14	5.30	7.66 (12 dpi)	Cyan (tumour)
UMAG_01432	acyltransferase fer5	524	3.85	−0.24	−4.39 (8 dpi)	Burlywood
UMAG_00133	1-alkyl-2-acetylglycero-phosphocholine esterase	20	3.66	−7.08	−2.58 (8 dpi)	Dark-green
UMAG_06404	peroxiredoxin	7147	3.61	0.35	1.50 (2 dpi)	Light-green (early biotrophy)
UMAG_02803	glycosyl hydrolases family 16 (GH16) domain-containing protein	68	3.59	−2.29	−6.04 (4 dpi)	Burlywood
UMAG_10260	peptide-methionine (S)-S-oxide reductase	352	3.26	−0.25	1.41 (2 dpi)	Cyan (tumour)
UMAG_03524	copper amine oxidase	29	3.22	2.76	5.29 (0.5 dpi)	Light-cyan
UMAG_05581	bifunctional cysteine synthase /	1183	3.20	1.28	2.58 (1 dpi)	Magenta (biotrophy)
UMAG_01232	O-acetylhomoserine aminocarboxypropyltransferase	1383	3.16	2.93	1.25 (0.5 dpi)	Light-cyan

## Data Availability

Raw and processed RNA-seq data are openly available at the GEO database under accession number GSE176292: https://www.ncbi.nlm.nih.gov/geo/query/acc.cgi?acc=GSE176292 (2 July 2021).

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
