# Peer review of "mRNA Inventory of Extracellular Vesicles from Ustilago maydis"

_jof, 2021, doi:10.3390/jof7070562_

Round 1

Reviewer 1 Report

In their study entitled “mRNA inventory of extracellular vesicles from Ustilago maydis” Kwon et al. report on mRNA molecules inside a fraction enriched for extracellular vesicles. These fraction was derived from U. maydis b-induced filaments grown in axenic culture. The authors collect extracellular vesicles by differential centrifugation and show that their preparations contain RNA, which becomes RNase sensitive only upon treatment with detergent. mRNA included in these preparations was enriched and sequenced. This sequencing data was compared to data obtained from b-induced filaments. In accordance to previously published data from mammalian cells but also from fungi, they find mRNA species of higher abundance in EV preparations compared to filaments. Of interest, these transcripts encode proteins, which cluster in certain GO terms. In addition, they show that transcripts upregulated during pathogenic development are often contained within EVs. Overall the authors make several interesting points and provide a valuable framework for future exosome research in U. maydis. This fungus could be an interesting model as it lacks the RNAi machinery and might therefore not secrete small regulatory RNA molecules as found in other systems. The paper merits publication in principle. However, I have some comments and concerns, which should be addressed by the authors.

  1. The major criterium used by the authors for EV enrichment is RNA, which becomes susceptible towards RNase treatment upon addition of detergent. To judge the quality of the preparations it would be better to also include combinations of protease and RNase as RNA inside of vesicles should be protease protected while RNA residing within extracellular mRNPs might well be protected by protein and thus could be degraded upon combining RNase and protease. I don’t think it is necessary to repeat any of the laborious experiments, but having a look at bioanalyzer profiles of extracellular preparations treated as stated above is important to understand if the transcriptome data provided represents mRNA encapsulated in EVs. At least the authors should better say that they have enriched for EVs rather then isolated or purified them (especially in the abstract section of the paper)
  2. Did the authors search for a common motif inside of mRNA species, which are overrepresented in the EV fraction. Selective sorting into EVs might require such a motif and the identification would improve the impact of the paper. There are already examples in the literature at least for RNA contained in mammalian EVs  (e.g. Kossinova et al. 2017, BBA; Joerger-Messerli et al. 2018, Cell transplant)
  3. Did the authors find any evidence for unspliced or alternatively spliced transcripts accumulating in EVs. There are several examples for alternative splicing in U. maydis (e.g. Freitag et al. 2012, Nature; Rodriguez-Kessler et al. 2011, Microbiological Research; Ho et al. 2007, BMC Genomics). It could be interesting if only one or the other splice variant becomes packaged in EVs. Such an analysis would also be a benefit to address the point mentioned in 2.
  4. It is interesting that the authors identify several mRNAs highly abundant in EVs (e.g. UMAG_02984). Did the authors delete corresponding genes to reveal their impact on virulence or filament formation? This might be out of the scope of the current story but could be interesting for the future.

Reviewer 2 Report

In this manuscript, Kwon et al. investigated the mRNA repertoire of extracellular vesicles (EVs) of the plant-pathogenic fungus Ustilago maydis. To analyze which impact mRNA transmitted via EVs to the host plant might have they took advantage of a laboratory strain capable to form filaments and thereby mimic infectious stages of U. maydis. Isolated EVs were analyzed by means of TEM and FM4-64 staining and were proven to protect the contained RNA.

Sequencing of poly(A)-enriched libraries of RNA from EVs and hyphal filaments 758 transcripts significantly enriched and 1189 transcripts depleted from EVs suggesting a selective loading of RNA into EVs. Interestingly, a very large proportion had full CDS coverage and represent full-length mRNAs. Shorter mRNAs encoding metabolic enzymes, oxidoreductases, and transcripts of previously identified effector genes were enriched within EVs,

The study is interesting and of importance in the field of plant pathogenicity and fungal/plant communication. It opens the door for future research addressing the function of mRNAs transmitted by EVs. The experimental work presented was carefully done.

Only a few points should be improved.

l. 15 plant-pathogen instead of plants-pathogen

l. 19-21 this sentence is hard to understand, perhaps split into two sentences

l. 53 [23] not in italics

l. 63 explain the term sRNA

l. 184 Table S5 mentioned before Tables S2, S3 and S4

l. 202 – 207 refer to Fig. 2a, b and c in the text.

Figure 3d show the numbers of genes enriched or depleted (758 and 1189) above the pie diagram.

Section 3.3 please mention the criteria why exactly these four genes have been chosen for transcript analysis and mention what is encoded by these genes.

References: Genus names should be written with the first letter capitalized, sub-species names first letter lower case. Also RNA instead of rna!

Round 2

Reviewer 1 Report

Thanks for addressing my points. As a hint for the future: I think many people are doing their protease protection experiments on 4°C, which still works fine.